# The Role of Temperate Phages in Bacterial Pathogenicity

**DOI:** 10.3390/microorganisms11030541

**Published:** 2023-02-21

**Authors:** Vimathi S. Gummalla, Yujie Zhang, Yen-Te Liao, Vivian C. H. Wu

**Affiliations:** Produce Safety and Microbiology Research Unit, U.S. Department of Agriculture, Agricultural Research Service, Western Regional Research Center, Albany, CA 94710, USA

**Keywords:** temperate phages, foodborne pathogens, horizontal gene transfer, virulence factors, antibiotic resistance genes, biofilm formation

## Abstract

Bacteriophages are viruses that infect bacteria and archaea and are classified as virulent or temperate phages based on their life cycles. A temperate phage, also known as a lysogenic phage, integrates its genomes into host bacterial chromosomes as a prophage. Previous studies have indicated that temperate phages are beneficial to their susceptible bacterial hosts by introducing additional genes to bacterial chromosomes, creating a mutually beneficial relationship. This article reviewed three primary ways temperate phages contribute to the bacterial pathogenicity of foodborne pathogens, including phage-mediated virulence gene transfer, antibiotic resistance gene mobilization, and biofilm formation. This study provides insights into mechanisms of phage–bacterium interactions in the context of foodborne pathogens and provokes new considerations for further research to avoid the potential of phage-mediated harmful gene transfer in agricultural environments.

## 1. Bacteriophages Influence Host Behavior

Bacteriophages (also known as phages) are viruses that infect bacteria and archaea. They are the most abundant biological entities on Earth, estimated at 10^31^, and are ubiquitous in different environments, such as oceans, lakes, and soil [1,2,3,4]. Phages are either virulent or temperate based on their life cycles [5,6,7]. Phage infections start with the recognition and binding to receptors on the surface of the host, which in this case, are bacterial cells. After injecting their DNA into bacteria, virulent and temperate phages proceed differently. Virulent phages (also known as strictly lytic phages) can only have a lytic cycle, where the phage hijacks host cell machinery to produce new phage particles and eventually lyse the host bacteria to release the infectious progenies. Temperate phages (also known as lysogenic phages) can initiate either the lytic or lysogenic cycle after infection. It is highly dependent on the surrounding environmental conditions, such as bacterial density and nutritional stress. In the lysogenic cycle, temperate phages integrate their genomes into the bacterial chromosomes, known as lysogenization, as a form of prophages, which can replicate along with the bacterial genome. Temperate phages can also enter the lytic cycle by carefully being cut, or excised, out of the host genome spontaneously or in response to external stresses via induction to release the prophages from the host genome, leading to host lysis [8,9]. The evolutionary purpose of lysogenic phages, their integration into the host genome, and the delayed start of the lytic cycle are multifaceted and, ultimately, serve to produce and spread more infectious phage particles across a bacterial population [6].

In contrast to virulent phages as the natural predators of bacteria, temperate phages are more favorable to bacterial evolution, particularly for bacterial pathogenicity, by naturally introducing new functional genes to the host genome through generalized (virulent and temperate phages) or specialized (temperate phages only) transduction [5]. Phage-mediated transfer of virulence genes and antibiotic resistance genes (ARG) to their bacterial hosts are highly associated with enhanced bacterial pathogenicity and fitness. In addition, prophage integration and excision from host chromosomes can also influence bacterial cellular properties, such as biofilm formation. The association between temperate phages with bacterial virulence and fitness reveals the potent nature of these phages contributing to the evolution of bacterial pathogenicity and nuance of our understanding of ‘temperate’ phages [10].

## 2. Profile and Genetic Characteristics of Prophages within Foodborne Pathogens

With the increasing amount of research using whole-genome sequencing technology to study foodborne pathogens, more and more prophages are found in bacterial genomes [11,12]. These genomes are known as poly-lysogenic, a term used to describe a bacterium with multiple prophages. The common pathogenic bacteria related to foodborne illnesses in the United States and the brief overviews of their respective prophage profiles are summarized here. They can be divided into two broad categories: Gram-negative bacteria and Gram-positive bacteria, due to the differences in the mechanisms of phage infections: phages that infect Gram-negative bacteria attach to extracellular targets such as lipopolysaccharides (LPS), O- and K- antigens, and outer-membrane proteins, while phages that infect Gram-positive bacteria target molecules on the peptidoglycan (PG) cell wall such as teichoic acids [13,14].

### 2.1. Gram-Negative Bacterial Pathogens

Several Gram-negative bacteria, including Shiga toxin-producing *Escherichia coli* (STEC), *Salmonella*, *Vibrio*, and *Campylobacter*, are highly associated with foodborne outbreaks in the United States and can cause various human diseases such as hemolytic-uremic syndrome (HUS), salmonellosis, cholera, and inflammatory bowel disease (IBD) [15,16,17,18,19]. The most infamous prophages of STEC pathogens are Stx1- and Stx2-converting phages, consisting of *stx* genes encoding the Shiga toxins [20]. The initial discovery regarding the toxin genes located on prophages instigated a more extensive search for prophages in the genomes of foodborne pathogens. A previous study conducted in our lab investigated the prophage profiles from 40 complete genomes of different STEC strains originally isolated from diverse sources, such as creek sediment and feces samples [20]. The results showed that about 8–22 different prophage sequences in each STEC genome were predicted, with 75.4% intact and inducible; among those prophages, the three most common prophages were genomically similar to Stx1- and Stx2-converting phages and the *Enterobacteria* phage lambda [20]. Within *Salmonella* pathogens, the analysis performed by Mottawea et al. displayed that prophage sequences were most prevalent in *S*. *Typhimurium*, with about nine prophages per isolate, but least prevalent in *S. Havana*, with about two prophages per isolate [21]. In addition, Wahl et al. found that there were an average of five prophages in one *S. enterica* isolate, making up about 3.52% of the total host genome [4]. Among those *S. enterica* prophages, the best-known prophage is P22, while the most common one is the lambdoid prophage Gifsy-2 [4,21]. Moreover, some prophages, such as Gifsy-2 prophages within two phylogenetically distinct strains (*Salmonella Typhimurium* ST313 strain D23580 and ST19 strain 4/74), had conservative nucleotide sequence regions with similar gene expression patterns, indicating that the sequences of *Salmonella* prophages are relatively conserved compared to that of other bacterial genera [4,22]. In *Vibrio* species, *Vibrio cholerae* and *Vibrio parahaemolyticus* are common human pathogens that are also poly-lysogenic [23]. Castillo et al. reported that prophages were frequently found in the genomes of *Vibrio* pathogens isolated from both human and marine animals. Prophage CTX𝜑, which carries two very different and harmful toxin genes: *ctxAB* encoding cholera toxins and *zot* encoding *Zonula occludens* toxins (Zot), was the most studied and relevant to this pathogen [23]. Moreover, the authors observed that many prophage-like elements in *Vibrio* samples contained at least one of the following functions: toxins (cholera and Zot toxins), antibiotic resistance, environmental adaptation, or metabolic enzymes. These findings likely suggest that temperate phages are highly associated with gene transfer across various *Vibrio* species from different sources [23,24]. *Campylobacter* prophages were also found to carry the *zot* gene encoding Zot toxins. Liu et al. showed that these Zot toxins expressed by *Campylobacter* prophages had very distinct amino acid sequences from those that had been previously examined in *V. cholerae* prophages [25]. However, Zhang et al. found that Zot toxins from *Campylobacter concisus* shared conserved elements with the *V. cholerae* Zot receptor-binding domain, indicating the mechanism that the Zot toxins from *Campylobacter* used to damage intestinal permeability might be similar to Zot from *Vibrio* [26]. These findings provide new evidence of prophage-mediated transfer of the *zot* gene to *Campylobacter*. However, their specific properties have yet to be determined. 

### 2.2. Gram-Positive Bacterial Pathogens

Certain Gram-positive bacteria, including *Listeria*, *Clostridium*, and *Staphylococcus*, can also cause severe human diseases—such as listeriosis, gas gangrene, and bloodstream infections (BSIs)—and hence contribute to foodborne outbreaks [27,28,29,30]. Prophages are present in most *Listeria* serotypes, but many have yet to be genetically sequenced and examined in detail. A previous study examined 90 *Listeria monocytogenes* isolates from food or food-related environments and found that 14.4% of the strains carried at least one prophage [31]. Uniquely, *Listeria* prophages have been involved in active lysogeny, where prophages regulate host gene expression while maintaining the lysogenic cycle [32,33,34]. As an example, prophage 𝜑10403S in *L. monocytogenes* strain 10403S integrates and excises out of the *comK* gene region, with a predicted function as a master transcription activator. At the lysogenic stage, the prophage inhibits *comK* expression to support the growth of *L. monocytogenes* during mammalian cell infection. On the other hand, prophage excision restores the function of the *comK* gene and, thus, acts as a regulatory switch for bacterial gene expression, unlike the prophage induction that results in the production of phage particles and subsequent bacterial host lysis [32]. *Clostridium perfringens* isolates are also poly-lysogenic because each strain can contain up to eight prophages, with *Clostridium phage vB CpeS-CP51* being the most abundant [35]. Two genomic characteristics of *C. perfringens*—14.1% of its accessory genome from prophages and low prevalence of CRISPR-Cas (clustered regularly interspersed short palindromic repeats-CRISPR-associated proteins) systems—indicate that its genome is highly adaptable for gene transfer. Specifically, a CRISPR-Cas system is an active defense mechanism used by bacteria to recognize and destroy bacteriophages. The system utilizes a Cas nuclease directed by non-coding RNA to the complementary genetic material of infecting bacteriophages and degrades it. While this mechanism is very common in bacteria, diverse phages, such as *C. perfringens* and *V. cholerae*, were found to have anti-CRISPR defense mechanisms [36]. Under the arms race between bacteria and phages, the combination of a low CRISPR-Cas prevalence and phage anti-CRISPR protection provided a great opportunity for prophage integration and survival. Altogether, these findings indicated a high possibility of gene transfer by temperate phages in this pathogen [35]. Most *Staphylococcus aureus* isolates contain 1–2 different prophages, mostly belonging to the 𝜑3 and MR11-like phages [37,38]. Currently, there have been increasing numbers of diverse prophage sequences identified within the genomes of *S. aureus* strains, resulting in the discovery of more poly-lysogenic *S. aureus* strains. Moreover, a previous study found that 𝜑Sa3-like prophages shared low genomic similarities with their structural genomic variations and were related to various bacterial virulence factors, attributed to the high diversity of virulent-related *S. aureus* prophages [39].

The wide distribution of prophages within the genomes of foodborne pathogens and their co-evolution with bacterial hosts play a crucial role in bacterial pathogenicity and fitness via phage–bacterial interactions. Regarding key characteristics of temperate phages as mobile genetic elements, there is a lack of a comprehensive understanding of how temperate phages transfer various genes to better the pathogenicity and fitness of foodborne pathogens. Horizontal gene transfer among foodborne pathogens via transformation (uptake of foreign genetic material) and conjugation (transfer of genetic material between bacterium by direct contact via a pilus) has been extensively studied [40,41]. Yet, there are gaps in phage-mediated gene transfer, known as transduction (transfer of genetic material between bacterium by a bacteriophage). Therefore, this review summarizes the effect of temperate phages on bacterial pathogenicity through horizontal gene transfer in the context of foodborne pathogens and raises concern for the resulting consequences to food safety and public health (Figure 1).

## 3. Virulence Factors

### 3.1. Prophages Carry Toxin Genes 

Toxin genes are one of the common virulence factors carried by prophages and can be transferred to other closely related bacteria via transduction (Table 1). For example, Stx1- and Stx2-converting phages in *E. coli* can be induced under diverse external stresses, such as EDTA and UV, to trigger the production of Shiga toxins and infectious phage particles and release them in the surrounding environment after bacterial lysis [20,42,43]. Subsequently, the induced phages carrying *stx* can infect non-pathogenic *Enterobacter* and *E. coli* to, in turn, make them toxigenic. The study conducted by Schmidt H. reported that the transduction frequency of the Stx1-converting phage was between 10^−3^–10^−5^ transductants per donor cell in vivo [44,45]. Another well-known example is the *Vibrio* CTX𝜑 prophage, which carries two toxin genes: *ctxAB* and *zot*. *Vibrio* cells can express both toxin genes regularly because the CTX𝜑 prophage has unique mechanisms for integration and replication that allow the phage and its toxin genes to persist longer in *Vibrio* [24]. Unlike most mobile genetic elements, the study conducted by Pant et al. revealed that the CTX𝜑 prophage can integrate into the *V. cholerae* genome irreversibly (permanently in the host genome) and initiate its own replication in this integrated form if it is tandem or flanked by a satellite phage RS1 [24]. This characterization demonstrated that the CTX𝜑 prophage could produce infectious phage particles (phage particles are secreted through host-specific pores in the membrane) and supply the host cytotoxicity with both cholera and Zot toxins at the same time [24,46]. The Zot toxin is also encoded by *Campylobacter* prophages. A previous study found 12 distinct *zot* genes identified in prophage sequences from nine different *Campylobacter* species/subspecies genomes; the genetic variations of *zot* genes within *Campylobacter* prophages increase diversity in the pool of toxin-associated *Campylobacter* prophages [25]. Among the 16 studied *C. perfringens* isolates, four toxin genes (*cloSI*, *cpe*, *nanH*, and *plc*) carried by four intact prophages (*Clostridium* phage *phiMMP01*, *Clostridium* phage *vbCpeS-CP51*, *Clostridium* phage *PhiS63*, and *Staphylococcus* phage *SP beta-like*) were detected [35]. In detail, the *plc* gene encodes α-toxins responsible for gas gangrene, the *cpe* gene encodes *Clostridium perfringens* enterotoxin responsible for human food poisoning, and the *nanH* (encoding sialidase) and *cloSI* (encoding α-clostripain) are minor toxins without a known specific pathogenic effect. Importantly, the *cpe* gene, previously found in the bacterial host chromosome or plasmids, was reported on the *Clostridium* prophage *phiMMP01* for the first time [35]. This finding provides a novel insight into the toxin gene transfer, mediated by a temperate phage, to other *Clostridium* strains, resulting in enhanced bacterial virulence.

In addition to the intraspecific transfer of toxin genes via prophages for these previously described bacterial species, phage-mediated toxin gene transfer has also been observed between different bacterial species. For example, *Staphylococcus*-specific phages that integrate within SaPI1 (*Staphylococcus aureus* pathogenicity island-1) can package surrounding virulence genes from the pathogenicity island, such as the shock toxin TSST-1 gene, through induction and further transfer these genes to another species *L. monocytogenes* at high frequencies [47]. One possible explanation for this phenomenon could be that the SaPI1 has secondary integration sites that allow the prophage to integrate into more places within a genome and are similar to the integration sequences in *L. monocytogenes* strains. The phenomenon has also been found in Stx-converting phages engaging in horizontal gene transfer events between *Shigella* and *E. coli* strains, resulting in the emergence and continuous evolution of human-pathogenic *Shigella* [48,49].

Overall, the bacterial hosts can produce a greater toxin level and become competitive against other surrounding pathogens by obtaining toxin genes from temperate phages. On the other hand, the emergence of bacterial pathogens with enhanced virulence poses a risk to human health through consuming contaminated foods. Therefore, a better understanding of the prevalence and mechanisms of toxin-carrying prophages is needed to take the proper steps to minimize food safety risks. 

**Table 1 microorganisms-11-00541-t001:** Temperate phage-mediated virulent gene (VG) and antibiotic-resistant gene (ARG) transfer.

Bacterial Host	Temperate Phages	VGs/ARGs	Toxin/Antibiotic Resistance	Transduction Efficiency	Recipient	References
*Escherichia coli*	Stx1- and Stx2- converting phages	*stx*	Shiga toxin	10^−3^ to 10^−5^ tru ^a^/cell	Non-pathogenic *Enterobacter* and *E. coli*	[20,44,45]
	*E. coli* prophage	*blaTEM*, *blaCTX-M*	β-lactams	NA ^b^	*E. coli*	[50]
	Stx phages 933W, 557, 312, and Cdt phage	*armA*	Aminoglycosides	NA	*E. coli*	[51]
	SUSP1, SUSP2	*kan*, *amp*	Kanamycin, ampicillin	NA	*E. coli*, *Bacillus* sp., soil bacteria	[52]
	Stx-converting prophage 933w	*tet*	Tetracycline		*E. coli*	[50,53]
*Salmonella*	Fels-2, *Enterobacteriaceae*	*oqxB_1*, *blaCTX-M*	Quinolones, β-lactams	NA	NA	[54]
	ES18	*amp*, *tet*, *cam*	Ampicillin, tetracycline, chloramphenicol	10^−8^, 10^−9^, 10^−7^ tru/pfu ^c^	*S.* Typhimurium	[55]
*Vibrio cholerae*	CTX𝜑	*ctxAB*, *zot*	Cholera toxin, Zonula occludens toxin	NA	*V. cholerae*	[23,56]
*Campylobacter concisus*	CON_phi2	*zot*	Zonula occludens toxin	NA	*C. concisus*	[25,26]
*Clostridium perfringens*	*Clostridium* phage *phiMMP01*, *vbCpeS-CP51*, *PhiS63*, *Staphylococcus* phage *SP beta-like*	*cloSI*, *cpe*, *nanH*, *plc*	α-toxins, *Clostridium perfringens* enterotoxin, sialidase, α-clostripain	NA	*C. perfringens*	[35]
*Staphylococcus aureus*	80α	*TSST-1*	Shock toxin TSST-1	10^−1^ tru/pfu 10^−1^ to 10^−6^ tru/pfu	*S. aureus* *L. monocytogenes*	[47]
	Staphylococcal phages	*cat*, *aadDE*, *msrA*	Chloramphenicol, aminoglycosides, macrolides	NA	*S. aureus*	[56]
	80α	*bla*, *tet*	Penicillin, tetracycline	10^−5^, 10^−6^ tru/pfu	*S. aureus*	[56,57]
	PDT17	*amp*, *cam*	Ampicillin, chloramphenicol	10^−8^ tru/pfu	*S.* Typhimurium	[55]
	Staphylococcal phages	*bla*, *fusB*	Penicillin, fusidic acid	NA	*S. aureus*	[56]
	80α	*str*	Streptomycin	10^−1^ tru/pfu	*L. monocytogenes*	[47]

^a^ Transduction unit. ^b^ No data available. ^c^ Plaque-forming units.

### 3.2. Prophages Increase Bacterial Adherence

Bacteria cells are flexible and must employ numerous strategies to quickly adapt to new adhering conditions and surfaces. Adhesion is highly related to the attachment and colonization of bacterial cells and plays a critical role in the initial step of bacterial infection. Several studies have indicated that the adherence abilities of bacteria were greatly influenced by the presence of temperate phages in the genome [58,59]. For example, the comparison of avian pathogenic *E. coli* (APEC) DE205B with and without prophage phiv205-1-infecting chicken embryo fibroblasts cells DF-1 indicated the adhesion abilities of the bacteria without the prophage significantly decreased from 3.3 × 10^6^ CFU/mL to 2.1 × 10^6^ CFU/mL compared to the bacteria with the prophage [60]. A later discovery by Wahl et al. demonstrated that the *gpE* gene on prophage SopE𝜑 could increase the adhesion of *S. enterica* to epithelial cells by encoding a putative tail-spike protein at cold temperatures [4,61]. In addition, prophage CJIE1 has been demonstrated to increase the adherence of host *C. jejuni* to human intestine cells INT-407; the *C. jejuni* isolates containing prophages had approximately 6-to-7-fold greater adherence than the isolates without prophages [62]. These findings reveal evidence of prophages positively associated with increased bacterial adherence, an early and necessary virulence factor for bacterial infection.

### 3.3. Release of Virulence Molecules Coordinated by Prophages

After virulence-associated genes brought by prophages are transcribed and translated to produce virulence molecules, these virulence molecules need to be released from their bacterial cells to enhance bacterial infection. One release pathway for virulence molecules involves bacterial cell lysis mediated by prophage induction [63]. For *Enterobacteriaceae* and *Salmonella* lysogens, toxins exit the cell along with phage particles because prophage induction has evolved to minimize the cost of cell death and maximize the spread of infectious phages [8]. When a temperate phage undergoes induction and enters the lytic cycle, the phage’s lytic proteins, such as lysins and hydrolases, can lyse the host cell, potentially releasing everything in the cell, including virulence molecules [14,64]. Subsequently, the virulence molecules can then diffuse to their target and cause damage through either a receptor-mediated or -independent manner [65,66]. 

The second pathway is mediated by outer membrane vesicles (OMV), which also can work with prophages [67,68]. OMVs are spherical, bilayer membrane structures directly derived from the outer membrane of Gram-negative bacterial cells. They contain various outer membrane proteins, periplasmic and cytoplasmic proteins, PG, LPS, DNA, RNA, and other enzymes [68]. The function of OMVs is still under investigation but they are likely to be involved in intercellular communication, DNA transfer, antibiotic resistance, and the release of virulence molecules. OMVs pinch off the outer membrane of Gram-negative bacteria but can also be released from the cells inside out alongside phage-induced lysis. Phage-induced lysis releases OMVs, which protect phage-encoded toxins against degradative enzymes. For example, in some STEC strains, the Stx2 toxin was carried within OMVs, which could render the toxin protection and subsequently internalized into human intestinal epithelial cells for intracellular toxin delivery [67]. Additionally, phage-induced lysis is also coordinated with OMV production. Pasqua et al. observed that more OMVs were produced when the lytic genes of *E. coli* K12 prophage DLP12 were not expressed (under the treatments of low pH, high osmolarity, and low temperature); this may result from the accumulated proteins, PG- and LPS- fragments, necessary to be transported out of the cell [68]. However, the general role of prophages in OMV production and the OMV-associated transport of virulence molecules is still not clear. 

Phage induction facilitates appropriate cell lysis and the production and release of virulence molecules, posing a potential risk to human health. Above are the most common mechanisms that temperate phages enhance bacterial virulence; uncovering the intricate layers behind temperate phages and bacterial virulence has truly just started. 

## 4. Antibiotic Resistance Genes

Antibiotic resistance occurs when bacteria mutate or acquire new genes, particularly ARGs, to protect the cells from the damage caused by these antimicrobial agents. The emergence of antibiotic-resistant pathogens, such as *Salmonella* and *E. coli* strains, may derive from the misuse and overuse of antibiotics targeting the pathogens in potentially contaminated raw meat, poultry, and vegetables [50,57,69]. There is an increasing number of studies showing that phages isolated from agricultural environments, such as animal fecal waste and human/animal-contaminated river water, were found to carry multiple ARGs [70,71]. Thus, understanding the mechanisms of how temperate phages mobilize ARGs is critical to improving treatments against resistant bacteria. Recently, the phage-mediated transmission of ARGs has been closely associated with numerous foodborne pathogens, including *Salmonella*, *Clostridium*, *Streptococcus*, and *Staphylococcus* (Table 1) [72]. There are three viable ways phages can facilitate the movement of ARGs across bacterial cells (Figure 1b). First, the temperate phage can carry ARGs and then introduce the genes to bacterial host genomes after phage integration. Second, during the induction of a temperate phage, the host cell lyses and releases an intact ARG-containing plasmid, which could enter a new bacterial cell through transformation. In some cases, an ARG-containing plasmid can mistakenly be packaged in a phage capsid and subsequently be transferred to a new bacterial cell by phage transduction [51]. Third, the temperate phage can collect ARGs from the bacterial chromosome by generalized or specialized transduction. The former is when the phage hydrolyzes host DNA and packages a fragment in a phage capsid for transfer [50,55]. The latter is when the phage mistakenly picks up an adjacent ARG from the host chromosome during prophage excision [73]. 

### 4.1. ARGs within the Prophage Genome

Previously, Larrañaga et al. indicated that phage proteins, such as capsid shell proteins, could provide ARGs better protection than plasmids or free DNA [69]. Furthermore, in a study regarding the distribution of antibiotic resistance genes, the authors found that two β-lactamase-resistant genes—*blaTEM* and *blaCTX-M*—encoding resistance to β-lactam antibiotics, such as penicillin, were discovered on *E. coli* phages isolated from various agricultural-related environments, including wastewater treatment sites, farm runoffs, and urban sewage [50]. Kondo et al. investigated the structural features of prophage elements containing ARGs in *E. coli* and ESKAPE pathogens (*Enterococcus faecium*, *Staphylococcus aureus*, *Klebsiella pneumoniae*, *Acinetobacter baumannii*, *Pseudomonas aeruginosa*, and *Enterobacter* spp.) which are strains that can evade commonly used antibiotics with their rapidly growing multi-drug-resistant properties [55]. Among these prophages, the phage sequence regions carrying an integron usually harbored three or more ARGs; however, 82.7% of the prophages without an integron contained less than three ARGs. The finding demonstrates that the presence of an integron enables temperate phages to encode more ARGs. Furthermore, ARGs are usually located at the end of the prophage region, such as phage integrase, likely because most essential structural genes of the phage are in the central area. As an example, the *Salmonella* phage Fels-2 and *Enterobacteriaceae* phage 186 contain a phage-derived integrase close to its ARGs *oqxB_1* (quinolone resistance) and *blaCTX-M-15_1* (β-lactamase resistance), respectively [55,74]. 

### 4.2. ARGs within Plasmids

Plasmids are extrachromosomal DNA within a cell that can replicate independently. Most plasmids are circular, but other qualities of plasmids like their copy number (number of copies of a plasmid in a host cell) and genomic sequences, are more variable. The investigation of phage-mediated gene transfer of plasmid-borne ARGs by Rodríguez-Rubio et al. was conducted by transducing *E. coli* plasmids with different copy numbers to recipient *E. coli* strains via Stx-converting phages. It showed that ARGs located on high-copy number *E. coli* plasmids, such as the *armA* gene (aminoglycoside resistance), were encapsulated up to 10,000 times more efficiently than those on low-copy plasmids [51]. Consistently, the study reported by Valero-Rello et al. demonstrated that small high-copy number plasmids were much more likely to be transduced by phage SPP1 than large low-copy number plasmids [75]. The authors found that the transduction efficiency of a low-copy number plasmid was about 4.6 times less than that of a high-copy number plasmid with the same amount of DNA and circularization rate. Furthermore, the presence of a sequence homologous to any region of a prophage in the plasmid of the same bacterial host increased transduction frequencies by more than 1000-fold. This evidence suggests that a plasmid harboring part of prophage sequences could be a signal for the transduction of that plasmid. This phenomenon was observed in *Bacillus subtillis* with phage SPP1 but has yet to be confirmed in foodborne pathogens. Sometimes temperate phages can exist as plasmids, also called phagemids, which can facilitate gene transfer too. For example, P1 phagemid derivatives harbor ARGs, such as *amp* and *mcr-1*, and can transduce those genes to *E. coli* strains [76]. Another phage–plasmid interaction that promotes ARG transfer is when a temperate phage initiates the lytic cycle and lyses the bacterial cell to release ARG-carrying plasmids. In most cases of the lytic process, plasmids are degraded by nucleases, but some can be released intact in the environment and become acquired by other bacteria, along with their ARGs [52,56,77]. Keen et al. discovered two *E. coli* phages SUSP1 and SUSP2 that successfully promoted the transformation of plasmids with ampicillin and kanamycin resistance [52]. Similarly, *S. aureus* phages have also been shown to promote plasmid-mediated gene transfer of ARGs [56]. The increasing relevance of interactions between phages and plasmids on ARG transfer has been indicated by researchers and should be further studied. 

### 4.3. ARGs within Bacterial Chromosomes

Additionally, when a prophage excises during induction, the host’s ARGs could be accidentally acquired and packaged with the phage genome [50,55]. Generalized transduction seems to occur more commonly in foodborne pathogens compared to specialized transduction. Schmieger et al. analyzed both P22-like temperate phages ES18 and PDT17 in *S. Typhimurium* DT104. They found that phage ES18 was highly associated with the transfer of tet (tetracycline-resistance), cam (chloramphenicol-resistance), and amp (ampicillin-resistance) genes. Phage PDT17 was closely related to *amp-* and *cam*-resistant genes [55]. Furthermore, phage ES18 could co-transduce multiple ARGs, creating a penta-resistant phenotype of *S. Typhimurium* [50]. Regarding *E. coli*, Colavecchio et al. noticed that induction of the Stx-converting prophage 933w in *E. coli* O157:H7 successfully transferred tetracycline-resistant cassettes to the laboratory strain *E. coli* K-12 [50,53]. Despite these observations, both *Salmonella* and *Enterococcus* bacteria were found to have low transduction frequencies of ARGs in general. Phage ES18 in *Salmonella* transduced *tet* at a rate of 10^−8^ transductants/pfu, while phage EGRM195 in *Enterococcus* transduced *tet* at a rate of 10^−8^–10^−9^ transductants/pfu. One possible explanation is that ARG-transfer events rely on a series of mistakes during phage excision. On the other hand, *S. aureus* pathogenicity islands (SaPIs), which contain many ARGs that confer resistance to streptomycin, fusidic acid, and penicillin, were found to be more susceptible to transduction than other parts of the host genome. Novick et al. found that SaPIs utilized phage satellites (small phages that help prophages replicate) to transfer these ARGs among *S. aureus* strains via specialized transduction [73].

The scenarios discussed above highlight the complexity and diversity of phage-mediated ARG transfer. Currently, viral metagenomics has been used to improve the screening of ARGs carried by phages for the relative ARG distribution in different environmental microbiomes [78,79]. For example, Moon et al. discovered that diverse ARGs, such as *blaHRV-1* and *blaHRM-1*, were encoded by infectious phages or prophages via metagenomic analysis [78]. The virome metagenomics conducted by Debroas et al. also indicated that ARGs were detected both in free viruses and prophages isolated from different environments, such as oceans, freshwater ecosystems, soil, and human guts [79]. Therefore, metagenomics and other advanced technology could improve the depth and accuracy of experimental design regarding the natural risk of phage-associated ARG transfer in future research.

## 5. Biofilm-Related Genes

Foodborne pathogens form biofilms not only for additional protection against environmental stress but also to increase the infectivity of their hosts [80,81]. The biofilm formation ability of bacteria is also affected by prophages in the bacterial host genomes [82]. There have been several studies conducted that address the relationship between temperate phages and biofilm formation. Although some studies found that a temperate phage integrated into a bacterial genome increased the biofilm capacity of the bacterial host, some studies suggested that prophage excision resulted in better biofilm formation. These two mechanisms are described in detail below.

### 5.1. Prophage Enhances Biofilm Formation

Concerning biofilm enhancement, the phiv205-1 prophage in APEC strain DE205B significantly increased the biofilm formation capability by approximately 52.38% compared to the wild-type strain without the prophage [60]. In addition to biofilm formation, prophage phiv205-1 also increased the colonization capacity of APEC strain DE205B for systemic infections because the phage carried a gene encoding putative endo-alpha-sialidase associated with colonization. Previous studies proposed three potential mechanisms that explain how prophages can increase biofilm formation: induction of the prophage causes extracellular DNA (eDNA) release and enhances biofilm production, the prophage encodes proteins that promote the formation of biofilms, or prophage excision activates motility operators (discussed in the next section) [60,83]. Further, Shah et al. found evidence of a temperate phage encoding biofilm-associated proteins in *Salmonella Typhimurium* strain [61]. The authors found that the *Salmonella* strain expressed the protein STM2699, encoded by the Fels-2 prophage of *S. Typhimurium*, to alter the bacterial cell surface for better biofilm formation on the surface of Caco-2 cells during long-term refrigeration storage at 5 °C. Another study found that deletion of prophage CTXφ from the *V. cholerae* strain decreased the bacterial biofilm formation by more than two times compared to the wild-type strain; the hypothesis for the decreased biofilm was that the expression of the *ctxAB* operon within prophage CTXφ also regulated the expression of primary biofilm-related genes, such as *vpsT* and *vpsR* (*Vibrio* polysaccharide) [84]. In addition, Yang et al. discovered that the prophage of *V. parahaemolyticus* encoding the gene *VpaChn25_0724* contributed to biofilm formation; the strain without the *VpaChn25_0724* gene showed a 1.5-fold decrease in biomass compared with wild-type strains after 24 h of bacterial growth. The findings indicated that the absence of the phage-encoded *VpaChn25_0724* gene disrupted the biofilm formation of the *V. parahaemolyticus* strain CHN25 [85].

### 5.2. Prophage Excision Leads to Greater Biofilm Formation

Alternatively, prophages can also increase bacterial biofilm formation by being excised out of the host genome without producing infectious phage particles and lysing the host cell. In *E. coli* K-12, prophage rac was the first defective prophage discovered, and excision of the phage could induce biofilm formation [86]. Liu et al. found that *E. coli* K-12 strain BW25113 without the prophage rac (mutant) formed 7.9-fold more biofilms at an early stage of biofilm formation (8 h) compared to the strain with the prophage (WT) [86]. A similar phenomenon was observed on the Stx1 prophage in the *E. coli* O157:H7 strain PA20 [87]. The Stx1 prophage integrated into the *mlrA* gene region, which encoded a transcription factor that indirectly regulated the expression of curli and amyloid proteins: both essential components of the *E. coli* biofilm matrix. Therefore, after the Stx1 prophage was excised, the *mlrA* gene was able to express to enhance biofilm formation. Accordingly, the author further confirmed that the *E. coli* O157:H7 strain PA20 with Stx1 prophages expressed fewer curli than the variant strain without Stx1 prophage at different temperatures (25, 30, and 37 °C). The findings suggest that prophage excision is a regulatory mechanism for primary biofilm-associated genes to enhance biofilm formation. 

## 6. Conclusions

Although knowledge regarding temperate phages barely scratches the surface, the current findings provide relevant evidence of the phages driving the evolution of common foodborne pathogens. The continuously occurring and widely spreading foodborne bacteria with enhanced pathogenicity and antibiotic resistance have become a serious food safety issue worldwide. Ultimately, temperate phages play vital roles in bacterial pathogenicity and fitness through horizontal gene transfer and could be a potential risk jeopardizing the existing antimicrobial intervention technologies. The ecological and epidemiological consequences of phages in spreading virulent genes and ARGs pose legitimate risks to human health. The contributing factors, such as antimicrobial interventions used in agricultural operations, should be scrutinized due to the complexity of phage–bacterial interactions within diverse environments. In addition, future studies are necessary to develop tools, such as CRISPR technology, to minimize tempting temperate phage-mediated gene transfer in agricultural environments.

## Figures and Tables

**Figure 1 microorganisms-11-00541-f001:**
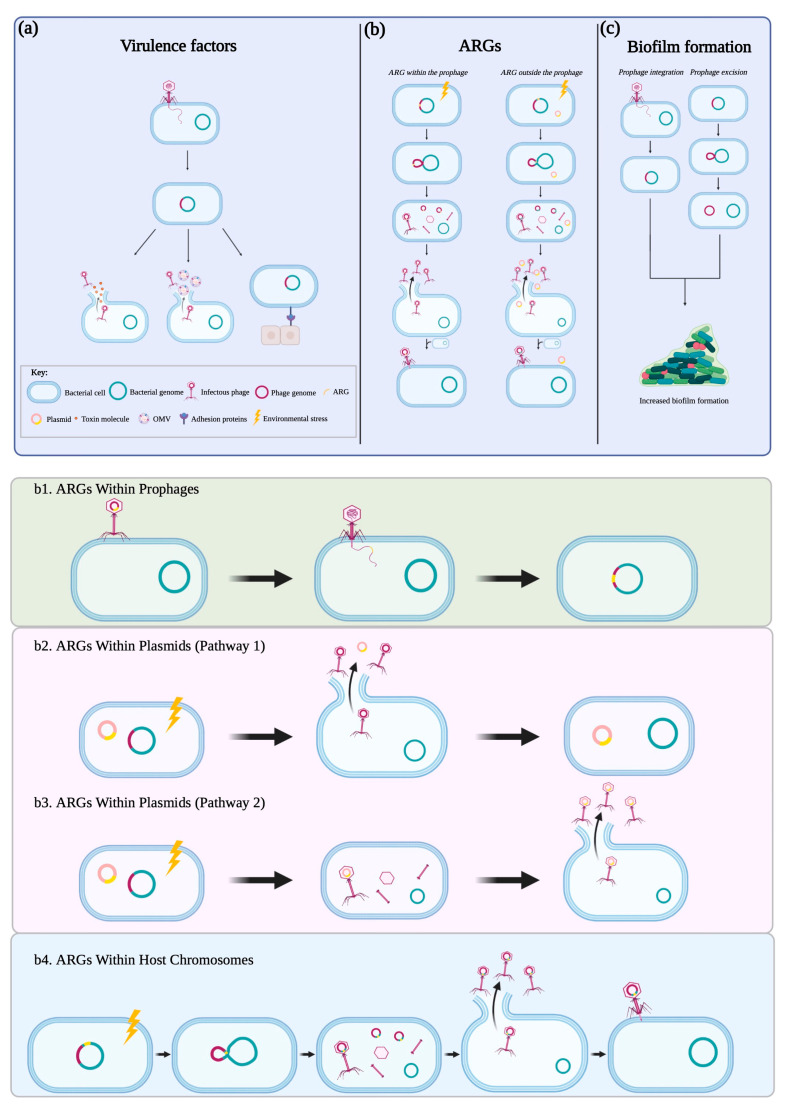
Temperate phages’ contributions to host bacterial pathogenicity. There are three major ways that temperate phages increase the pathogenicity of their host. (**a**) Firstly, temperate phages can transfer the virulence genes encoding for toxins and adhesion proteins to their hosts (leftmost panel). (**b**) Secondly, temperate phages can transfer ARGs originally located within the phage genome or from exogenous sources (host chromosome and plasmids), contributing to the emergence of antibiotic-resistant strains (middle panel). The mechanisms of phage-mediated transfer of ARGs are further demonstrated in detail (bottom panel): (**b1**) The temperate phage can carry ARGs and then introduce the genes to a new bacterial host genome after phage integration. (**b2**) During the induction of a temperate phage, the host cell lyses and releases an intact ARG-containing plasmid, which can enter a new bacterial cell through transformation. (**b3**) In some cases, an ARG-containing plasmid can mistakenly be packaged in a phage capsid and subsequently transferred to a new bacterial cell by phage transduction. (**b4**) The temperate phage can collect ARGs from the bacterial chromosome by generalized or specialized transduction. (**c**) Finally, gene transfer by temperate phages can promote bacterial biofilm formation through either their integration into or excision out of the host genome, increasing the host’s resistance to antimicrobial treatments (rightmost panel). Abbreviations: OMVs, outer membrane vesicles; ARG, antibiotic resistance genes.

## Data Availability

Not applicable.

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
