# Peer review of "The Role of Temperate Phages in Bacterial Pathogenicity"

_microorganisms, 2023, doi:10.3390/microorganisms11030541_

Round 1

Reviewer 1 Report

This is a review and attempted synthesis of literature addressing temperate phages in bacterial pathogenicity.  This has been a well recognized aspect of bacterial pathogen evolution for decades.  However, this always value in revisiting such topics periodically to capture new examples and new concepts.  While this review cites a great deal of more recent literature, the organization and overall theme seem poorly developed.  An early section deals with food-borne pathogens while subsequent sections are related to specific categories of genes.  One approach, i.e. by types of pathogens, or by types of genes should be followed.  The devotion of one section to foodborne pathogens and the occasional references to foodborne pathogens implies a focus to this article that is never properly introduced or explained.  In addition, it is important to keep in mind that review articles have an important role in education and hence should be accurate in their use of terminology.  In this review there are a number of essentially incorrect statements.  Some examples follow:

lines 2-28.  Phage are virulent or temperate by nature they do not become one or the other after genome injection.

line 34.  The term transduction does not refer to incorporation of a phage genome into a bacterium.  This is lysogenization.  The incorporation in this way of other genes of possible value to the host is usually referred to as lysogenic conversion.

lines 139-140 and elsewhere.  Transformation is stated to be plasmid-mediated here however this is more a feature of a given species than of a particular plasmid.  In addition transformation with linear chromosomal fragments is common in some species.  Conjugation is stated as being uptake of free DNA.  This is incorrect.

lines 283-290.  This is an opportunity to define for the reader the terms generalized transduction, specialized transduction, and lysogenic conversion.

lines 285-7.  Again, confusion over what conjugation is.  In this instance the authors state that plasmid DNA released by cell lysis can enter another cell by conjugation.  This is incorrect.

Lines 289-290.  The authors invoke specialized transduction as a mechanism for transfer of adjacent antibiotic resistance genes but do not provide a reference.  

Lines 346-348.  Again, the authors invoke specialized transduction involvement in ARG transfer.  Two references are provided.  One is a review article that did not appear to support the statement.  Primary literature should typically be cited in a review article unless indicated otherwise.  The other reference specifically refers to generalized transduction in the abstract as the mechanism for ARG transfer in that paper.

lines 314-5.  Not all plasmids are small.

Author Response

Reviewer 1

This is a review and attempted synthesis of literature addressing temperate phages in bacterial pathogenicity. This has been a well recognized aspect of bacterial pathogen evolution for decades. However, this always value in revisiting such topics periodically to capture new examples and new concepts. While this review cites great deal of more recent literature, the organization and overall theme seem poorly developed. An early section deals with foodborne pathogens while subsequent sections are related to specific categories of genes. One approach, i.e. by types of pathogens, or by types of genes should be followed. The devotion of one section to foodborne pathogens and the occasional references to foodborne pathogens implies a focus to this article that is never properly introduced or explained.

Response: We appreciate the reviewer’s comment and have adjusted a paragraph to make a better transition between the foodborne pathogen and gene transfer section and focused more on information related to the purpose of this review manuscript (lines 148-160).

In addition, it is important to keep in mind that review articles have an important role in education and hence should be accurate in their use of terminology. In this review there are a number of essentially incorrect statements. Some examples follow:

lines 2-28. Phage are virulent or temperate by nature they do not become one or the other after genome injection.

Response: We appreciate the reviewer’s comment and have reworded the sentence in the revised manuscript (lines 24-28).

line 34. The term transduction does not refer to incorporation of a phage genome into a bacterium. This is lysogenization. The incorporation in this way of other genes of possible value to the host is usually referred to as lysogenic conversion.

Response: We appreciate the reviewer’s comment and have updated the sentence in the revised manuscript (line 34).

lines 139-140 and elsewhere. Transformation is stated to be plasmid-mediated here however this is more a feature of a given species than of a particular plasmid. In addition transformation with linear chromosomal fragments is common in some species. Conjugation is stated as being uptake of free DNA. This is incorrect.

Response: We appreciate the reviewer’s comment and have rewritten the sentence in the revised manuscript (lines 153-157).

lines 283-290. This is an opportunity to define for the reader the terms generalized transduction, specialized transduction, and lysogenic conversion.

Response: We appreciate the reviewer’s comment and have added definitions to make the terms clearer (lines 308-312).

lines 285-7. Again, confusion over what conjugation is. In this instance the authors state that plasmid DNA released by cell lysis can enter another cell by conjugation. This is incorrect.

Response: We appreciate the reviewer’s comment and have updated the sentence in the revised manuscript (line 308).

Lines 289-290. The authors invoke specialized transduction as a mechanism for transfer of adjacent antibiotic resistance genes but do not provide a reference.

Response: We appreciate the reviewer’s comment and have added a reference to provide an example of specialized transduction (line 312, line 383-388).

Lines 346-348. Again, the authors invoke specialized transduction involvement in ARG transfer. Two references are provided. One is a review article that did not appear to support the statement. Primary literature should typically be cited in a review article unless indicated otherwise. The other reference specifically refers to generalized transduction in the abstract as the mechanism for ARG transfer in that paper.

Response: We appreciate the reviewer’s comment and have made a distinction between generalized and specialized transduction in this section (lines 367-388). We have also added the primary research article relevant to the findings discussed in our article (lines 383-388) 

lines 314-5. Not all plasmids are small.

Response: We appreciate the reviewer’s comment and have updated the sentence in the revised manuscript (line 336).

Reviewer 2 Report

This review paper easily follow the content and good explanation. However, some points in this review should clarify to make it clear as followed.

1)     The authors clearly explained how to transfer virulence genes or antibiotic resistance genes to other bacteria. However, this review focus the pathogenicity gene transfer only in the same species. The authors should not ignore the possible way of pathogenicity-genes transfer to other species such as Shiga toxin (Shigella sp and E. coli). 

2)     The authors mention about “the molecular mechanisms of phage-bacterium interactions and provokes new considerations for avoiding phage-mediated harmful gene transfer during the food production and processing chain” in the abstract. However, there are not good explanation in this review about this topic.

3)     On the Table 1 (Page 7), it is not quite clear what is the number of 214. In addition, the table is difficult to see the information between bacterial host. The authors should edit to make it easily see.

Author Response

Reviewer 2

This review paper easily follow the content and good explanation. However, some points in this review should clarify to make it clear as followed.

  • The authors clearly explained how to transfer virulence genes or antibiotic resistance genes to other bacteria. However, this review focus the pathogenicity gene transfer only in the same species. The authors should not ignore the possible way of pathogenicity-genes transfer to other species such as Shiga toxin (Shigella sp and E. coli).

Response: We appreciate the reviewer’s comment and have reworded lines 213-224 to emphasize the example we originally provided about inter-species gene transfer. We have also added an additional example about gene transfer between Shigella and E. coli to this paragraph (line 222-224).

  • The authors mention about “the molecular mechanisms of phage-bacterium interactions and provokes new considerations for avoiding phage-mediated harmful gene transfer during the food production and processing chain” in the abstract. However, there are not good explanation in this review about this topic.

Response: We appreciate the reviewer’s comment and have reworded the abstract to align with the content and purpose of this manuscript more clearly (lines 15-17). We have also added two examples of phage-mediated ARG transfer in agriculture settings (line 295-298).

  • On the Table 1 (Page 7), it is not quite clear what is the number of 214. In addition, the table is difficult to see the information between bacterial host. The authors should edit to make it easily see.

Response: We appreciate the reviewer’s comment and have updated the Table 1 to better distinct the information for each bacterial host. The missing line number 232 refers to the content of Table 1.

Reviewer 3 Report

Bacteriophages, viruses that infect bacteria, are able to replicate following two cycles: the lytic and lysogenic cycle. In the lytic cycle, in a short time, they can be replicated in new virions which are released causing cell rupture, while in the lysogenic cycle, the virus (temperate phage) integrates into the bacterial genome and will silently replicate with the latter, waking up only in stressful situations. Some studies report that temperate phages can benefit their bacterial hosts by introducing additional genes into their bacterial chromosomes. This review reports how temperate phages are involved in the bacterial pathogenicity of foodborne pathogens, including phage-mediated virulence gene transfer, antibiotic resistance gene mobilization, and biofilm formation.

The subject matter is not new, although, to date, many mechanisms are not yet known. Therefore, the manuscript contains interesting research ideas and in-depth insights for operators in the sector and deserves to be read.

The purpose of the article has been correctly defined.

Literature selection is appropriate throughout the manuscript.

On the other hand, some modifications are needed, as follows.

Paragraphs throughout the manuscript should be numbered.

Enter a list with the main acronyms reported in the text and their respective full names.

Line 126. Clustered regularly interspaced short palindromic repeats (CRISPR) are widespread in bacterial genomes and act as an active defense mechanism to protect against bacteriophage infection. Therefore, for the benefit of readers, I suggest inserting a short paragraph on current knowledge of the CRISPR system and making the acronym explicit.

Author Response

Reviewer 3

Bacteriophages, viruses that infect bacteria, are able to replicate following two cycles: the lytic and lysogenic cycle. In the lytic cycle, in a short time, they can be replicated in new virions which are released causing cell rupture, while in the lysogenic cycle, the virus(temperate phage) integrates into the bacterial genome and will silently replicate with the latter, waking up only in stressful situations. Some studies report that temperate phages can benefit their bacterial hosts by introducing additional genes into their bacterial chromosomes. This review reports how temperate phages are involved in the bacterial pathogenicity of foodborne pathogens, including phage-mediated virulence gene transfer, antibiotic resistance gene mobilization, and biofilm formation.

The subject matter is not new, although, to date, many mechanisms are not yet known. Therefore, the manuscript contains interesting research ideas and in-depth insights for operators in the sector and deserves to be read.

The purpose of the article has been correctly defined.

Literature selection is appropriate throughout the manuscript.

On the other hand, some modifications are needed, as follows.

Paragraphs throughout the manuscript should be numbered.

Response: We appreciate the reviewer’s comment and have updated the section numbers in the revised manuscript.

Enter a list with the main acronyms reported in the text and their respective full names.

Response: We appreciate the reviewer’s comment and have added a list of main acronyms at the end of the revised manuscript.

Line 126. Clustered regularly interspaced short palindromic repeats(CRISPR) are widespread in bacterial genomes and act as an active defense mechanism to protect against bacteriophage infection. Therefore, for the benefit of readers, I suggest inserting a short paragraph on current knowledge of the CRISPR system and making the acronym explicit.

Response: We appreciate the reviewer’s comment and have added information about the CRISPR system and its relation to phage-bacterial relationships in the revised manuscript (line 128-139).